# A Contactless Glucose Solution Concentration Measurement System Based on Improved High Accurate FMCW Radar Algorithm

**DOI:** 10.3390/s22114126

**Published:** 2022-05-29

**Authors:** Chengjie Liu, Yuan Du, Li Du

**Affiliations:** School of Electronic Science and Engineering, Nanjing University, Nanjing 210000, China; cj_liu@smail.nju.edu.cn

**Keywords:** FMCW radar, wavelet transform, deep learning algorithm, PSD, sparrow search algorithm, convolutional neural network

## Abstract

To reduce the pain and the probability of cross-infection caused by the invasive blood glucose testing instruments, the ex vivo glucose measurement is of high significance. The electrical property of blood varies with the density of the glucose, which can be sensed by measuring its reflected coefficient in millimeter-wave. In this article, we built a contactless glucose solution concentration measurement system based on 77-GHz FMCW radar. Several preliminary signal processing algorithms are cascaded with a deep neural network to improve the accuracy of glucose solution concentration measurement. Our experiment shows that the resolution of this ex vivo glucose measurement can achieve up to 0.1 mg/mL.

## 1. Introduction

With the improvement of the population’s living standards, more and more people are suffering from diabetes. According to an investigation by WHO, the number of people with diabetes rose from 108 million in 1980 to 422 million in 2014. In other words, 1 in 11 people has diabetes. Between 2000 and 2016, there was a 5% increase in premature mortality from diabetes [1]. This will not only burden the national medical finance but also affect the population’s health level. The traditional invasive method of blood glucose monitoring hurts the patient and may cause cross–infection. Non–invasive blood glucose monitoring can avoid the above problems and has become a promising glucose measurement method for growing diabetes.

To achieve the non–contact measurement of the blood glucose concentration, a direct method would be measuring the dielectric constant of the blood [2]. As blood glucose concentration in human blood will affect EM properties more significantly than other chemicals, similar to vitamins and metabolites [3], the dielectric constant will be a direct reflection of the body’s blood glucose concentration. There are many ways to monitor blood glucose, such as surface plasmon resonance, electromagnetic sensing, and optical coherence tomography [4,5,6]. Among many methods of measuring dielectric constant, millimeter–wave is one of the most suitable methodologies as it can penetrate the skin easily and is sensitive to small changes in glucose concentration [7,8,9,10,11].

There are many methods to use millimeter-wave to measure the dielectric constant and electrical properties of the human blood sample with different glucose concentrations or aqueous glucose solution, such as using a vector network analyzer to directly measure the dielectric constant and the loss tangent of the solution [12], measuring the S parameters of blood glucose [13], and using resonant perturbation method to measure the changes of resonant frequency [14]. The methods above require bulky experimental apparatus and measuring kits which are not suitable for real-time and portable blood glucose monitoring. The millimeter radar can be reduced into a single chip that enables real-time, lightweight monitoring of blood glucose in diabetic patients.

In this paper, we used a Frequency Modulated Continuous Wave (FMCW) radar integrated into a single chip as a portable device to continuously measure the glucose aqueous solution concentration. The FMCW radar sends an electromagnetic wave to the target and receives the reflected signal from the object. The received signal of the FMCW radar is related to the object’s location, speed, and composition [15], the results will be unreliable if the equipment and the object are not fixed firmly. Except for this, the disturbance of some other electromagnetic waves from outside will also be received by the antenna of the radar, which will also affect the result. To fix the problems mentioned above, a 3D printing device was used to fix the position of the radar and the container, and a microwave absorbing sponge was used to cover the device to lessen the disturbance from the outside. The same container combined with an injector for solution replacement was used to prevent the influence caused by the tiny mismatch of the containers’ surface and to make sure the same volume of solution is under monitoring.

According to the research [12,16], the difference in the dielectric constant will lead to a change in the amplitude and phase of the radar echo signal. In this way, the solution of different glucose concentrations can be classified by the amplitude and phase of the radar echo signal. Except for the traditional radar signal processing methods [17,18], there are also some other signal processing methods to analyze the data, such as continuous wavelet transform (CWT), empirical mode decomposition (EMD), and discrete wavelet transform (DWT). EMD and DWT can be used to decompose the signal into different components which can show the composition of the signal and denoise the signal [19,20], and they have been used to analyze biosignals such as brain–death EEG data and Phonocardiogram signals [21,22]. They were used in this experiment to remove the high–frequency signal, sich as thermal noise and external electromagnetic interference, from the original data. As shown in the following sections, DWT was chosen to denoise the signal in this experiment as it can retain the signal’s energy at the low–frequency part where the main signal locates but EMD cannot. Nevertheless, the denoised signal will still be processed by FFT and PSD, which cannot show a clear relationship between the received signal and the solution’s concentration under different external environments.

CWT has been used in many fields to analyze different signals and proved to be useful when analyzing 1D signals such as heart sounds [23,24]. Compared to the above signal processing methods, which characterize the power or amplitude of the signal at one frequency, the CWT will turn the 1D signal into a 3D matrix which can show a clearer relation between time–domain and frequency–domain and can be used as the input of a convolutional neural network. As RF sensing or FMCW radar combined with a Neural Network has been proven to be able to identify different metals or glucose solutions of different concentrations [25,26], accordingly, the application of CWT will be combined with a neural network to investigate the deeper correlation between the concentration of the glucose solution and the character of the radar echo signal and improve the ability of classification even when the measuring environment is different. The main contributions of this paper are as follows:A proper experiment platform was built for measuring the concentration of the glucose solution. The experiment platform minimized interference to make sure the signal can characterize the electrical properties of the solution.The signal’s power analyzed by PSD was found to be a monotonic increasing function by the increase in the glucose concentration which accords with the theoretical result and the experiment result of other researchers [7,27]. As the radar’s power on status will influence the radar working status, meaning the fitting curve of every round of experiment cannot be confirmed, it is hard to measure the concentration of the glucose solution by only one fitting curve.An algorithm consisting of CWT and Neural Network is applied to classify the solution by glucose concentration in a deeper dimension which can solve the problem mentioned that the fitted curve cannot be confirmed, mentioned above.


## 2. Physical Principle

The electromagnetic wave can be divided into two parts: one whose electric field direction is vertical to the incident plane (s wave) and the other whose electric field direction is parallel to the incident plane (*p* wave). As the antennas of the radar that we used are vertically polarized, the vertical direction is the direction of the feed, that is to say, the incident wave is made up of an s wave. The following theoretical analysis focus on the s wave. The amplitude relation between reflected signal and incident signal is derivable from the Maxwell wave equation, and the reflected amplitude relation for s wave is shown in Equation (1), where $\theta_{i}$ means the incident angle, $\theta_{t}$ means transmitting angle, $\varepsilon_{1}$, $\mu_{1}$ separately means the dielectric constant and the magnetic permeability of the incident medium and $\varepsilon_{2}$, $\mu_{2}$ separately means the dielectric constant and the magnetic permeability of the transmission medium.
(1)$${\frac{\left|E_{R}\right|}{\left|E_{0}\right|}}_{⊥}=\left|\frac{\sin\left(\theta_{i}{\;-\;\theta }_{t}\right)}{\sin\left(\theta_{\;i}{+\theta }_{t}\right)}\right|=-\frac{\sqrt{\varepsilon_{2}\mu_{2}}{\cos\theta }_{t}-\sqrt{\varepsilon_{1}\mu_{1}}{\cos\theta }_{i}}{\sqrt{\varepsilon_{2}\mu_{2}}{\cos\theta }_{t}+\sqrt{\varepsilon_{1}\mu_{1}}{\cos\theta }_{i}}$$
(2)$$\frac{\partial {\frac{\left|E_{R}\right|}{\left|E_{0}\right|}}_{⊥}}{\partial \theta_{t}}=-\frac{\sin\left({2\theta }_{i}\right)}{\sin^{2}\left(\theta_{i}{+\theta }_{t}\right)}\;(\theta_{t}<\theta_{i})\;$$

As $\mu_{1}$ and $\mu_{2}$ are almost equal with each other, the relation between $\theta_{i}$ and $\theta_{t}$, which correlates with the dielectric constant, is shown in Equation (3).
(3)$$\frac{\sin\left(\theta_{i}\right)}{\sin\left(\theta_{t}\right)}=\sqrt{\frac{\varepsilon_{2}\mu_{2}}{\varepsilon_{1}\mu_{1}}}\approx \sqrt{\frac{\varepsilon_{2}}{\varepsilon_{1}}}\;$$

From Equation (2), if $\theta_{t}$ becomes smaller, which means the dielectric constant of the solution $\varepsilon_{2}$ grows bigger, the amplitude of the reflected signal of the s wave will increase. 

The antenna will not only receive the directly reflected signal, but also the transmitted signal reflected by the metal plate at the back. When considering the transmitted signal, it is necessary to confirm the attenuation of the signal when propagating in solution. The composition of the complex dielectric constant and the definition of loss tangent (tan(δ)) is shown in Equation (4), and the wave vector is shown in Equation (5).
(4)$$\{\begin{array}{c} {\varepsilon =\varepsilon }_{0}-i\frac{\sigma }{\omega }=\varepsilon^{\prime }+\varepsilon^{\prime \prime }\\ \tan\left(\delta \right)=\frac{\left|\varepsilon^{\prime \prime }\right|}{\varepsilon^{\prime }} \end{array}\;$$
(5)$$\{\begin{array}{c} k=\beta -i\alpha \;\\ \beta =\omega \sqrt{\mu \varepsilon }\sqrt{\frac{1}{2}(\sqrt{1+\frac{\sigma^{2}}{\varepsilon^{2}\omega^{2}}}+1)}\\ \alpha =\omega \sqrt{\mu \varepsilon }\sqrt{\frac{1}{2}(\sqrt{1+\frac{\sigma^{2}}{\varepsilon^{2}\omega^{2}}}-1)} \end{array}\;$$

The imaginary part of the complex dielectric constant represents the attenuation degree when the electromagnetic wave transmits into the solution as shown in Equation (6).
(6)$$E\left(r,t\right){=E}_{0}e^{-\mathrm{ikr}}{=E}_{0}e^{-\alpha r}e^{i\left(\omega t-\beta r\right)}\;$$

According to previous experimental research [7], the dielectric constant increases, and the loss tangent decreases with the increase in the solution’s glucose concentration in the range of 0.7 mg/mL to 1.2 mg/mL. Assuming the received signal is basically made up of the signal directly reflected from the front interface of the solution and the signal penetrating the solution and reflected by the metal plate, the amplitude of the received signal will be greater when the solution’s glucose concentration increases.

## 3. Working Environment

FMCW radar can send electromagnetic waves towards the target by the transmitted antenna (Tx). The electromagnetic wave is then reflected by the target, returns to the radar, and is received by the receive antenna (Rx). In this paper, we used an FMCW radar. It can deliver frequency modulated continuous waves whose frequency can vary linearly from a minimum to a maximum. A mixer after the Rx can multiply the Rx signal with the Tx signal which can generate a signal with two frequencies: the sum and the difference of RX signal frequency and TX signal frequency. The output of the multiplier will pass through a low pass filter and then be a signal with only one low frequency, which is the difference between the Rx signal frequency and Tx signal frequency. The frequency will represent the distance between the radar and target, which is caused by the time delay caused by the wave transmitting time during the whole course.

We used an FMCW radar named IWR1443 [28] which has three Tx and four Rx. It also has an MCU to modify the firmware and the method of data transmission. The 1D FFT result of the raw data can be read through a serial port and then transfer the FFT result back to the raw data by the IFFT algorithm. The specific process is shown in Figure 1a.

As mentioned above, 3D printing equipment was used to fix the devices’ position with microwave absorbing sponge cover around, as shown in Figure 1b, and equipped the equipment with an injector to change the solution more conveniently, as shown in Figure 1c.

## 4. Preliminary Signal Processing

To validate the correctness of the theoretical derivation, FFT and PSD were used to analyze the amplitude and power of the signal. The specific operation and related experiment data are shown in this section.

### 4.1. Received Signal Preprocessing

FFT and PSD can transform the time–domain signal into a frequency–domain signal which can show the amplitude and power of a fixed frequency. The zero–padding method was used to lessen the effect caused by the fence effect and therefore there is a quality–improved FFT or PSD result that can show a clearer relationship between the signal and the solution’s concentration. The raw data is processed by IDFT first, as shown in Equation (7), where *N* means the length of the raw data, c means the concentration of the glucose solution, and t means the number of the Tx antenna. Additionally, then using the zero–padding method, the first *N* values are set as the value of the corresponding sequence and the last 15 * *N* values are set as zero, as shown in Equation (8).
(7)$$X_{c}^{t}\left(n\right)=\frac{1}{N}\overset{N}{\underset{k=1}{\sum }}X_{c}^{t}\left(k\right){*e}^{\frac{j*2\Pi *\mathrm{nk}}{N}}\;$$
(8)$$s_{c}^{t}\left(n\right)=\{\begin{array}{c} x_{c}^{t}\left(n\right),\;\;\;\;\;1\leq n\leq N,\\ 0,\;\;\;\;\;N+1\leq n\leq 16N, \end{array}$$

After finding the result of the raw signal and the processed signal by zero–padding, the signal can be re–FFT, and the FFT result is named *S*(*k*). The PSD result can be found by *S*(*k*) as shown in Equation (9).
(9)$$P_{c}^{t}\left(k\right)=\frac{1}{N}{\left|S_{c}^{t}\left(k\right)\right|}^{2}$$

In what follows, the power of the received signal can be characterized by the sum of the *P*(*k*) as shown in Equation (10), and it can also be characterized by the peak value of *P*(*k*) which represents the signal of a certain frequency with the highest energy.
(10)$$E_{c}^{t}=\overset{16*N}{\underset{k=1}{\sum }}P_{c}^{t}\left(k\right)$$

As the result can be affected by the thermal noise and the irrelevant signal from the surrounding environment, concentrated in the high–frequency part, this can easily affect the result of Equation (10). Some methods were used to reduce the influence of the thermal noise which are introduced as follows. As the thermal noise concentrates in the high–frequency part and can be removed by a low pass filter, it is important to define the low pass point. Two kinds of methods were used to fix the problem above.

### 4.2. Empirical Mode Decomposition (EMD)

Empirical Mode Decomposition in conjunction with a Hilbert spectral transform, together called Hilbert–Huang Transform, is ideally suited to extract essential components which are characteristic of the underlying biological or physiological processes [29,30].

According to the principle of the EMD, the radar beat signal *x*(*n*) can be recognized as a combination of many different components as shown in Equation (11) where *C_i_*(*n*) means the ith component of the signal, *r*(*n*) means the residual part of the signal, and *P* means the number of signal components we expect.
(11)$$x_{c}^{t}\left(n\right)=\overset{P}{\underset{i=1}{\sum }}C_{i,c}^{t}\left(n\right){+r}_{c}^{t}\left(n\right)$$

After processing the *x*(*n*) by the EMD method, many signal components with different frequencies can be extracted from the original signal as shown in Figure 2a. The frequency of a component is defined by the zero–crossing points. The f_0_, which is the number of the zeros crossing point of *x*(*n*), is defined as a standard to differentiate the high–frequency components and low–frequency components. In this experiment, if the frequency of the *i*th component f_i_ is higher than or equals to three times f_0_, the ith component will be defined as a high–frequency component, and the converse is defined as a low–frequency component. After having defined the components as two parts, the high–frequency components will be removed from the signal to decline the thermal noise of high frequency in the signal.

### 4.3. Discrete Wavelet Transform (DWT)

Discrete Wavelet Transform (DWT) is a method derived from Continuous Wavelet Transform (CWT), which can be used to remove noise in the signal. It has been used in the field of image processing and has achieved good results [31]. The result of DWT can characterize the signal at different levels and can also well retain the peak and abrupt parts required in the original signal which are the main characteristics of analysis. The raw data was decomposed into three levels, and the fundamental wave is ‘db10’. The time–domain signal after being denoised by DWT is shown in Figure 2b.

After decomposing the raw signal, a denoised signal can be restructured by the coefficient from the result of DWT, and the frequency domain of the processed signal is shown in Figure 2c. It is clear to see that the time domain of the processed signal can accord with the trend of the original signal precisely, and the frequency domain of the de–noised signal is not only able to retain the information in the low–frequency region but also can filter the noise in the high–frequency domain.

From Figure 2c, the signal after being denoised by EMD and the original signal are much different in the low–frequency part of the frequency domain, and the signal after being denoised by DWT nearly coincides with the original signal in the low–frequency part of the frequency domain which is an important part to characterize the power of the signal.

### 4.4. Result of Analyzing PSD

To prove the repeatability of the experiment, the solution is measured five times. In this part, all the measurements were completed once from 0.69–1.08 mg/mL in one round and then repeated in the same way a further four times. The concentration of the solution is determined by the quality of added sugar, which is weighed by a precision electronic scale and the volume of the added water which is measured by a measuring cylinder. Figure 3a shows the time domain signal of different TX. Figure 3b shows the PSD result of the signal in the low frequency. The power of the signal is calculated by Equation (10), and the power of the signal of different glucose concentrations is shown in Figure 3c. Figure 3c shows that the power of the signal will increase as the glucose concentration increases.

After having demonstrated that this kind of method is promising for measuring the solution glucose concentration, it is also promising to measure other glucose solutions by the function obtained by linear fitting, whose concentration is not measured in the previous experiment. The method of linear fitting is the least square method, shown in Equation (12), where $\overset{-}{x}$ means the average value of $x_{i}$ and $\overset{-}{y}$ means the average value of $y_{i}$.
(12)$$\begin{array}{cc} \{ & \begin{array}{c} \hat{y}=k\hat{x}+b\\ k=\frac{\sum_{i=1}^{n}(x_{i}-\overset{-}{x})(y_{i}-\overset{-}{y})}{\sum_{i=1}^{n}{\left(x_{i}-\overset{-}{x}\right)}^{2}}\\ b=(\overset{-}{y}-k\overset{-}{x}) \end{array} \end{array}$$

After the fitted curve is obtained, the glucose concentration of the solution will be roughly extrapolated from the measured echo energy.

However, a problem will occur when fitting the curve in that the power of the external environment, such as temperature, will change the radar’s working status. This means that a fitting curve would only work before the radar turns off. Another set of experiments was performed in the same process, and the fitting curve is shown in Figure 3c. It is obvious to see that the slope and intercept of these two curves are different, which means that the curve will not be a reference to measure the concentration of the glucose solution after power off/on. In addition to the problem mentioned above, the nonlinear variation in the range from 1.03 mg/mL to 1.08 mg/mL will also lead to inaccurate concentration extrapolation using the fitted curve, and the use of higher order fitting curves can lead to overfitting, a situation that is not expected to happen.

## 5. Deep Learning Method

To solve the problem mentioned in Section 4, and considering that there are some correlations between the character of the radar echo signal and the concentration of glucose solution, which is quite hard to discover, the machine learning method was used. The whole progress includes CWT processing, data augmentation, Neural Network training, and the SSA finding an optimal hyperparameter setting, as shown in Figure 4a.

### 5.1. Datasets Partition and Augmentation

The input data were collected randomly and under different power–on statuses. There were 1000 effective sets of data collected; 85% were set as a training–set, 10% were set as a validation–set, and 5% were set as a test–set. After that, data in the training set were processed by adding white noise to expand the data set to solve the problem brought by the shortage of the data set.

### 5.2. Neural Network and Input Format

#### 5.2.1. Input Format

As the convolutional neural network is powerful when processing the 3D data, the original data needed to be processed into a 3D form. As the Continuous Wavelet Transform (CWT) can not only process the original time–domain complex signal into a 3D form, but also can carry more information to be analyzed, it is promising to be used as the input of the deep learning network. As the time–domain signal is complex–valued, the result of the CWT is a 3D matrix, where the first page is the CWT for the positive scales (analytic part or counterclockwise component) and the second page is the CWT for the negative scales (anti–analytic part or clockwise component) [32]. As the network cannot analyze complex–valued data, the data is restructured: let the first page be the real part of the CWT for the positive scales, the second page be the imaginary part of the CWT for the positive scales, the third page be the real part of the CWT for the negative scales, and the fourth page be the imaginary part of the CWT for the negative scales. The result X(ω, t, i), where *ω* means the angular frequency, *t* means the time, and *i* means the number of the layer, is then processed by zero–mean normalization. The Stochastic Gradient Descent Momentum (sgdm) was used as the optimizer to improve the convergence speed of the network. In addition, to save computing time, the length of the collected data is 64 instead of 256 or 512.

#### 5.2.2. LSTM Layer

LSTM is used when analyzing the data as a sequence, and it can greatly avoid vanishing gradients caused by recurrent neural network (RNN) [33,34]. It takes the result of the convolution kernel as the input, and it has been used in visual recognition and description [35]. LSTM is also used in this neural network to improve its classification ability of it [36].

In summary, ResNet18 was used as the convolutional part of the neural network, and two bi–LSTM layers were used behind the convolutional part. The network is used as a classification.

### 5.3. Sparrow Search Algorithm

The Sparrow Search Algorithm is a kind of optimizing algorithm which can find an optimal solution within a certain range [37]. This algorithm was used in this paper to find a relatively better hyperparameter setting so that the classification ability of the neural network can be improved.

In this experiment, the accuracy evaluated by the validation–set is set as the result of the function. The hyperparameters were set as the variations of the function, such as the size of the convolutional kernel, the stride of the convolutional kernel, and the size of the bi–LSTM kernel. After computing for multiple rounds, the result can relatively improve the classification ability of the neural network, and the training progress comparison between the default hyperparameter and the improved hyperparameter is shown in Figure 4b.

It is clear to see that the classifying ability of the Neural Network was improved by SSA. The accuracy of the validation set rose from 82% to 95%.

## 6. Discussion

The result of PSD has shown that it is promising to measure the concentration of the glucose solution. The PSD result processed from the de–noised time–domain signal shows that there is a positive proportional relation between the power of the reflected signal and the concentration of the glucose solution. The results of the experiment and the theoretical derivation agree with one another, and the results of the experiment also agree with the conclusion drawn from another research group.

As the behavior of the MOSFET will be infected by the environment when power is on, the linear function will be invalid after the radar is restarted. It was also quite difficult to ascertain the relation between the power of the reflected signal and the character of the reflected signal and then confirm the glucose concentration of the solution.

To prevent the infection of the power–on status, a machine learning method was used to figure out a deeper correlation between the character of the reflected signal and the glucose concentration of the solution. The CWT was also used to analyze the signal in both the time and frequency domains, and it can also be taken as the input of the CNN. The combination of CNN and bi–LSTM enabled the classification ability of the algorithm even when the data was collected after the radar was restarted, and SSA improved the classification ability further.

On the other hand, the classifier neural network can only be valid under a fixed glucose concentration range, and it will be invalid when the glucose concentration is not included in the test data. However, the regression neural network may have the ability to measure the glucose concentration which is not included in the test data and then improve the resolution of the measurement system. Applying the regression neural network is also part of our future research. The neural network with SSA also needs a huge amount of data to train the network and improve the classification accuracy, which is very time–consuming. The frame of the neural network also needs to be specialized to adapt either the input form of the neural network or the character of the input data.

The measuring range of the system is from 0.69 mg/mL to 1.08 mg/mL, which covers the normal human body’s blood–glucose concentration. In comparison with normal human blood–glucose variation, 0.7 mg/mL to 1.1 mg/mL, the final measurement resolution reached 0.05 mg/mL which means the system can measure the blood–glucose concentration level. The limit of detection and the limit of quantification are still needed to improve to reach a wider measurement range.

## 7. Conclusions

In this work, we developed an experiment device based on FMCW radar for distinguishing solution glucose concentration and solved the problem of inaccurate measurement due to different working statuses by the deep learning method. Firstly, we use FMCW radar to distinguish the glucose solutions of 0.69 mg/mL, 0.81 mg/mL, 0.91 mg/mL, 1.03 mg/mL, and 1.08 mg/mL concentrations successfully by the PSD result of the received signal, which has proven our ability to measure the solution concentration by FMCW radar. As the radar working status would be influenced when the power–on environment is different, in turn affecting the measuring result, a machine learning algorithm was used to ascertain the hard–to–find correlation between the glucose concentration and the character of the reflected signal.

The system has the following advantages:Low power consumption—the result of the Neural Network can be transplanted to the MCU, and the system is nearly made up of a radar circuit and an MCU;High accuracy—the resolution of the system is up to 0.1 mg/mL on average;Real–Time—the speed of collecting data is at the microsecond level, the speed of signal processing and calculating can be completed in a few seconds;Small Volume—the core module can be reduced to 5 cm × 5 cm × 2 cm;Remote sensing—the system has potential to be upgraded to monitor blood–glucose concentrations.

## Figures and Tables

**Figure 1 sensors-22-04126-f001:**
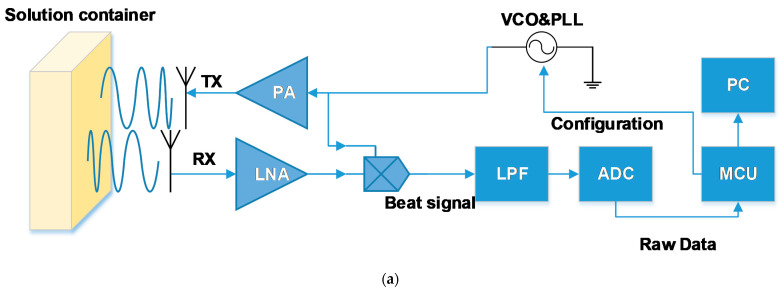
FMCW radar system: (**a**) specific data transfer flow chart; (**b**) the experiment device fixed by 3D printing equipment and covered by microwave absorbing sponge; (**c**) the device equipped with an injector.

**Figure 2 sensors-22-04126-f002:**
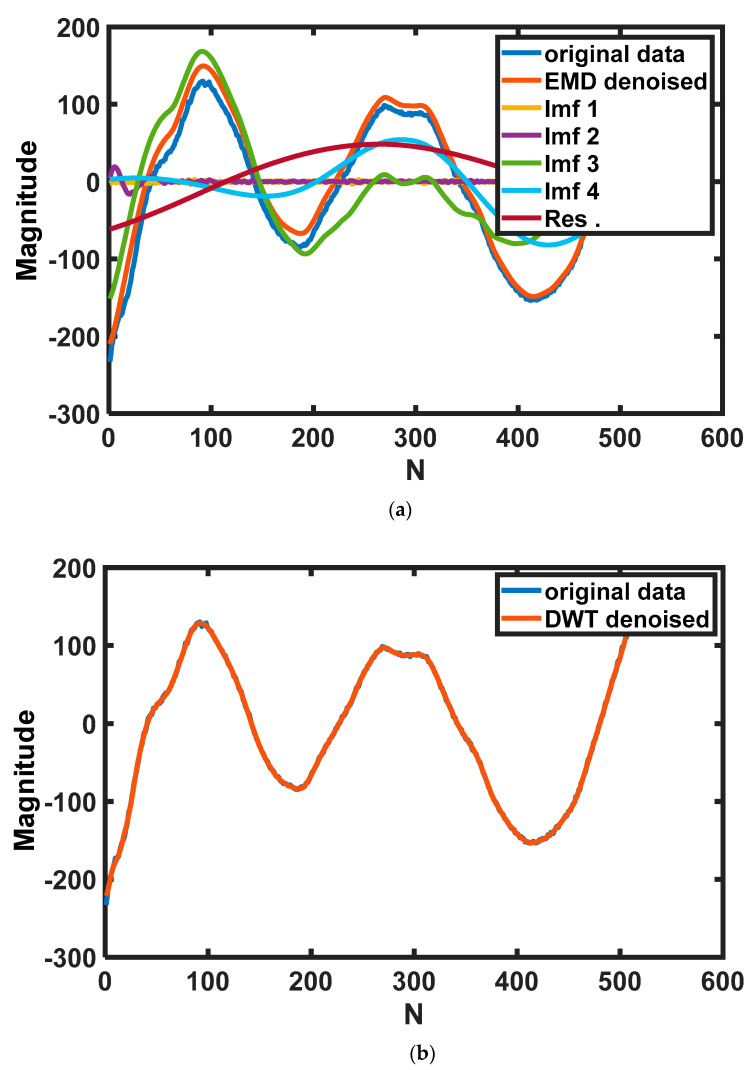
Using different methods to reduce thermal noise: (**a**) components of different frequencies extracted by EMD algorithm; (**b**) time–domain signal after denoised by DWT; (**c**) comparison of the result of EMD and result of DWT and raw data in the frequency domain.

**Figure 3 sensors-22-04126-f003:**
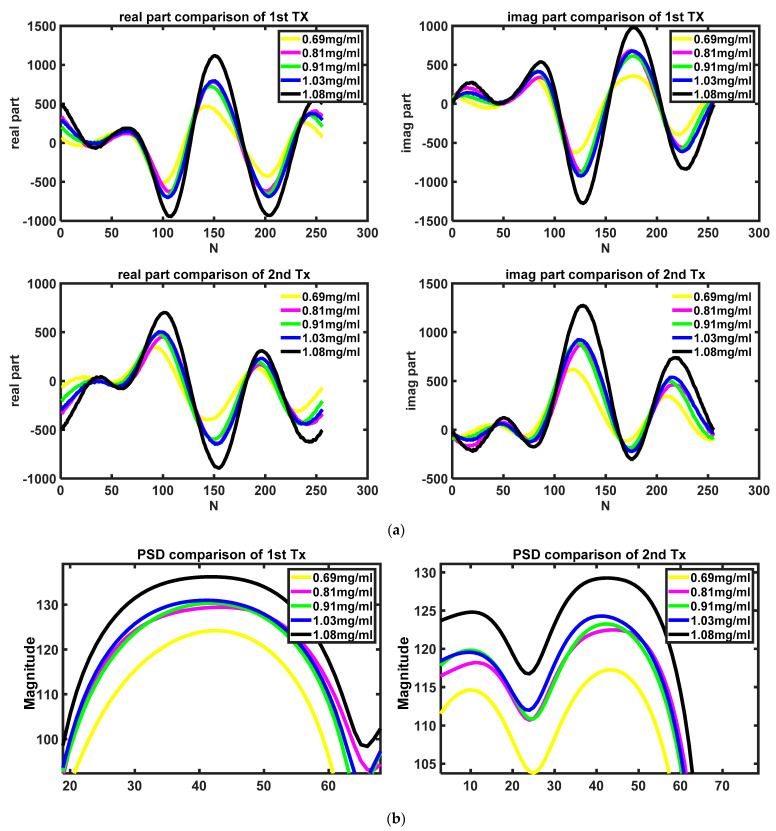
The result of the measurement: (**a**) time–domain signal of different concentrations and different TX; (**b**) PSD result of different concentrations in low–frequency part; (**c**) sum of PSD under different environments.

**Figure 4 sensors-22-04126-f004:**
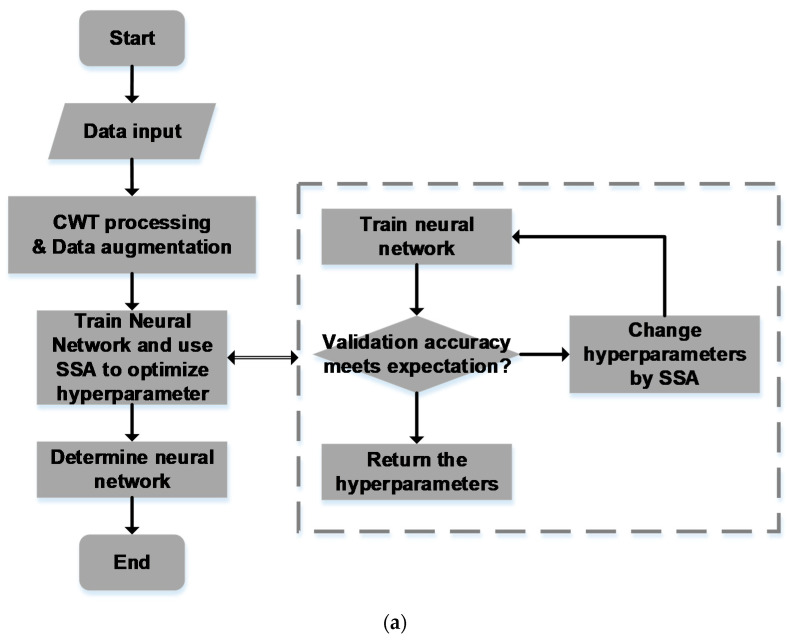
The deep learning method: (**a**) the progress of the whole deep learning algorithm with SSA; (**b**) the training progress comparison.

## Data Availability

Not applicable.

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
