# Peer review of "A Contactless Glucose Solution Concentration Measurement System Based on Improved High Accurate FMCW Radar Algorithm"

_sensors, 2022, doi:10.3390/s22114126_

Round 1

Reviewer 1 Report

  1. The authors do not show the specificity of this method. The authors should indicate whether the isomers like fructose or galactose may affect the detection of glucose or not.
  2. To this Reviewer, it is not completely clear what limit of detection (LOD), and limit of quantization (LOQ), exactly are. A reference to their definition (or one sentence for a better explanation) may be helpful for the non-expert reader. Since the resolution of the device is very important. This should be discussed and data should be presented. 
  3. In fact, to this Reviewer, it is not clear whether the Authors think that the sensor may be the basic component for a future device for noninvasive glycemia monitoring. If yes, certainly the experiments have done so far, though promising, are somehow preliminary, since experiments have not been done with blood. There is no information about the other different components of blood and their effect on the measured data you have presented.

Author Response

Response to Reviewer 1 Comments

1.The authors do not show the specificity of this method. The authors should indicate whether the isomers like fructose or galactose may affect the detection of glucose or not.

response1. Thank you for your question. As the frequency of EM waves emitted by millimeter wave is concentrated in GHz which means the EM waves can less scatter and go deeperr into the tissue to reach regions with sufficient blood
concentration, yielding more accurate glucose readings. Compared to other devices, radar is lighter and can be made more intergrated for real-time diabetic blood glucose concentration monitoring. As the isomers like fructose or galactose are not resident substances in the blood, our preliminary research doesn't take them into account.

2.To this Reviewer, it is not completely clear what limit of detection (LOD), and limit of quantization (LOQ), exactly are. A reference to their definition (or one sentence for a better explanation) may be helpful for the non-expert reader. Since the resolution of the device is very important. This should be discussed and data should be presented. 

response2.Thank you for your notice, the reference and the related discussion will be added into the revised manuscript.

3.In fact, to this Reviewer, it is not clear whether the Authors think that the sensor may be the basic component for a future device for noninvasive glycemia monitoring. If yes, certainly the experiments have done so far, though promising, are somehow preliminary, since experiments have not been done with blood. There is no information about the other different components of blood and their effect on the measured data you have presented.

response3.Thank you for your question. We do think this system can be the basic component for a future device and this article is about our preliminary experiments about proving the feasibility of using neural network for measuring human blood glucose concentration.  Any chemical compound will of course cause the EM properties of the blood. But according to an article (reference has been added in the revised manuscript), blood glucose concentration in human blood will affect EM properties more significantly than other chemicals, similar to vitamins and metabolites.

Reviewer 2 Report

This paper talked about an experiment device base on Frequency Modulated Continuous Wave radar for contactless glucose measurement. Authors also used a machine learning algorithm was to figure out the hard-to-find correlation between the glucose concentration and the character of the reflected signal. Unfortunately, I am not an expert in mathematic algorithm development. But, the detection detection range is good, the method for contactless, painless method is definitely needed for glucose measurement, the references are also up-to-date.

1) The author names and affiliations are missing in page 1.

2) The Y-axis label in Figure 2 and part of figure 3 is missing.

3) Talking about the sensor. From sensor performance perspective, what's the selectivity of detecting primary analyte over interference species? Or the sensor will just detecting any chemical compound? This is important because glucose is coexisting with many chemicals in blood.

4) what's the fitting curve from glucose measurement results? From Figure3, it doesn't looks like linear. What's your equation for predicting an unknown concentration of glucose?

Author Response

Response to Reviewer 2 Comments

1) The author names and affiliations are missing in page 1.

response1. Thank you for your notice, the author names and affiliations are added 

2) The Y-axis label in Figure 2 and part of figure 3 is missing.

response2. Thank you for your notice, the missing labels are added

3)Talking about the sensor. From sensor performance perspective, what's the selectivity of detecting primary analyte over interference species? Or the sensor will just detecting any chemical compound? This is important because glucose is coexisting with many chemicals in blood.

response3. Thank you for your question. Any chemical compound will of course cause the EM properties of the blood. But according to an article (reference has been added in the revised manuscript), blood glucose concentration in human blood will affect EM properties more significantly than other chemicals, similar to vitamins and metabolites.

4) what's the fitting curve from glucose measurement results? From Figure3, it doesn't looks like linear. What's your equation for predicting an unknown concentration of glucose?

response4. Thank you for your question. The fitting curve is a method for roughly extrapolate the glucose concentratin. After getting the fitting curve, the glucose concentration of the solution will be roughly extrapolated from the measured echo energy.But as you said, the variation is not linear which will cause the extrapolation is imprecise, and the use of higher order fitting curves can lead to overfitting. The problem above also drives the application of neural network.

Round 2

Reviewer 1 Report

Accept

Reviewer 2 Report

Even this is preliminary experimental data with many questions not yet addressed. This is still a very interesting and novel non-invasive method for glucose monitoring.